# Tackling the First COVID-19 Wave at the Cape Town Hospital of Hope: Why Was It Such a Positive Experience for Staff?

**DOI:** 10.3390/healthcare11070981

**Published:** 2023-03-29

**Authors:** Steve Reid, Mitan Nana, Theo Abrahams, Nadia Hussey, Ronit Okun-Netter, Tasleem Ras, Klaus von Pressentin

**Affiliations:** 1Primary Health Care Directorate, University of Cape Town, Cape Town 7700, South Africa; 2Dean’s Office, Faculty of Health Sciences, University of Cape Town, Cape Town 7700, South Africa; 3Western Cape Department of Health, Cape Town 8000, South Africa; 4Division of Family Medicine, University of Cape Town, Cape Town 7700, South Africa

**Keywords:** COVID-19, intermediate care facility, field hospital, psychological resilience, professional burnout, health team, interdisciplinary, health care team, culture, organizational, leadership

## Abstract

**Background**: In contrast to alarming reports of exhaustion and burnout amongst healthcare workers in the first wave of the COVID-19 pandemic, we noticed surprisingly positive staff experiences of working in a COVID-19 field hospital in South Africa. The 862-bed “Hospital of Hope” was established at the Cape Town International Convention Centre specifically to cope with the effects of the first wave of the COVID-19 pandemic in Cape Town. **Methods**: We aimed to systematically describe and assess the effects on staff and the local health system. A cross-sectional descriptive study design was employed using mixed methods including record reviews and interviews with key informants. **Results**: Quantitative results confirmed high job satisfaction and low staff infection rates. The emerging themes from the qualitative data are grouped around a “bull’s eye” of the common purpose of person-centeredness, from both patient and staff perspectives, and include staff safety and support, rapid communication, continuous learning and adaptability, underpinned by excellent teamwork. The explanations for the positive feedback included good disaster planning, adequate resources, and an extraordinary responsiveness to the need. **Conclusions**: The “Hospital of Hope” staff experience produced valuable lessons for designing and managing routine health services outside of a disaster. The adaptability and responsiveness of the facility and its staff were largely a product of the unprecedented nature of the pandemic, but such approaches could benefit routine health services enormously, as individual hospitals and health facilities realize their place in a system that is “more than the sum of its parts”.

## 1. Introduction and Background

An 862-bed COVID-19 field hospital, the “Hospital of Hope” (HoH), was established at the Cape Town International Convention Centre (CTICC) as an intermediate care facility specifically to cope with the effects of the Coronavirus disease 2019 (COVID-19) pandemic in Cape Town, South Africa. The temporary HoH was commissioned by the Western Cape Provincial Government on 1 April 2020 to meet the anticipated pressure on the acute hospital services in the city during the imminent first wave of the COVID-19 pandemic. The overall aim of the hospital was “to offer hope by delivering high quality, efficient inpatient care in response to the needs in the Cape Town metropole, while ensuring the safety and positive growth of staff.”

By 14 August 2020, medical care had been provided to 1502 patients, referred from other healthcare facilities in and around Cape Town. The 862-bed HoH facility accepted patient admissions for only a 10-week period (8 June 2021–14 August 2021), which saw 303 transfers out of the facility to higher levels of care and 81 in-patient deaths. The inpatient beds were organized into three main patient categories, namely, those patients diagnosed with COVID-19 requiring inpatient oxygen therapy, those patients diagnosed with COVID-19 that complicated their underlying chronic conditions, and those patients with COVID-19 deemed suitable for palliative care based on the Association of Palliative Care Practitioners of South Africa (PALPRAC) guidelines [1].

Owing to the severe stress, high emotional load, long working hours, concern over being infected or infecting loved ones, lack of adequate support in the working environment and lack of effective supportive treatment (especially during the first wave of the pandemic), the World Health Organization (WHO) identified healthcare workers (HCWs) as being at particular risk of developing burnout and fatigue because of working with COVID-19 patients [2]. A systemic review examining data from North America, Asia and Europe demonstrated that the severe acute respiratory syndrome (SARS), Middle East respiratory syndrome (MERS) and COVID-19 outbreaks had a substantial impact on the mental health of healthcare workers. [3] The study found that the top three most frequently reported mental health concerns included general health concerns (62.5% frequency), fear of contracting the virus (43.7%) and lack of sleep (37.9%). Other reported outcomes included psychological distress, anxiety, post-traumatic stress disorder, depression, burnout, stigmatization and somatization. As expected, healthcare workers in Asian countries, the epicenter of the previous MERS/SARS epidemics, did better than their Western counterparts during the COVID-19 pandemic, albeit only slightly. Moreover, a Turkish study found that physicians working in a “pandemic hospital” faced legal concerns and violence besides the diagnosis and treatment stress during the pandemic [4].

Stress, anxiety and depression-related disorders may be regarded as “normal” emotional reactions to a pandemic. Furthermore, field hospitals require highly qualified professionals to be able to act in even more extreme situations. A study conducted in Romania showed that 76% of employees at a COVID-19 field hospital had a high level of emotional exhaustion and depersonalization, and a low rate of personal achievement [5]. In Brazil, which experienced one of the highest rates of mortality among nurses, the main factors contributing to the deterioration in the mental health of healthcare workers were cited as poor working conditions, work overload and “the feeling of impotence against a new and highly contagious disease” [6]. High levels of emotional exhaustion in healthcare workers were reported in studies in France [7], Italy [8,9] and Spain [10]. Among the Spanish and Italian studies, emotional exhaustion and anxiety attacks were the main symptoms.

Yet, other field hospitals reported exactly the opposite. An Australian study [11] described positive aspects of working during the COVID-19 pandemic such as job security, redeployment to areas that were commensurate with skills, competence and experience, improved working relationships and teamwork, and a sense of purpose and contributing to something greater. Some study participants even reported “an improvement in mental health” owing to these factors [7]. While healthcare workers stationed at a field hospital in Brazil reported negative feelings owing to the fear of the unknown, working without protocols or protocols not underpinned by well-established scientific evidence, the use of PPE for prolonged periods, and communication and environmental difficulties, they also reported positive feelings associated with a display of empathy, teamwork and collaboration in a time of pandemic and overcrowding in healthcare services [12].

A scoping review of health worker experiences during the COVID-19 pandemic, published in 2022 [13], identified a lack of studies investigating HCWs other than doctors and nurses, HCWs in non-hospital settings, and HCWs in low- and lower middle-income countries. One qualitative study at Nightingale Hospital in Manchester reported that doctors had an overwhelmingly positive experience [14]. There was consistent mention of a strong team; in particular, the feeling of being individually valued within a flattened hierarchy. Staff well-being and education were also regularly mentioned and helped contribute to this overall feeling. When asked what they would take forward, doctors focused on the importance of a strong team that values multidisciplinary working. However, the hospital was not without challenges, with processes changing from one shift to the next and leading to potential errors. In addition, system issues such as medication and documentation led to a sometimes chaotic work environment.

Anecdotal feedback from the HoH team indicated that the experience of working at the Hospital of Hope was extraordinarily positive. The research question was “why was the staff experience at the hospital so positive?” We aimed to describe and assess the staff experiences and effects on the health system of the temporary HoH during the height of the first wave of the COVID-19 pandemic in Cape Town.

## 2. Methods

### 2.1. Study Design

A cross-sectional descriptive study design was employed using mixed methods including record reviews of existing documents, and interviews with key informants. A health systems perspective was taken as the conceptual framework for the study, using the WHO “building blocks” approach [15].

### 2.2. Study Setting

The population of the Cape Town metropole is served by both public and private sector health services. The majority of the 4,602,248 population have access to basic municipal services (service coverage of 96.4% for water, 94% for refuse removal, 93.7% for electricity, 90.9% for sanitation and 77.2% for housing). The public sector consists of 126 primary healthcare facilities, which drain to several district, regional and tertiary hospitals. These acute hospitals at regional and tertiary levels have high care and intensive care facilities available, including ventilators and specialist staffing. The two tertiary hospitals (Tygerberg and Groote Schuur hospitals) drain the whole Western Cape provincial public sector service.

### 2.3. Study Population

The HoH staff members consisted of 463 clinical staff (52 medical officers, 9 medical specialists, 360 nurses led by 8 operational managers, 11 physiotherapists, 4 dieticians, 8 pharmacists, 10 post-basic pharmacists, 4 radiographers and 3 social workers) and 210 non-clinical staff (administrative, supply chain, porters, housekeeping and catering staff), as well as an operational management team. During the commissioning of the HoH, 722 posts were approved and almost 400 of these were filled in record time. At first, there were insufficient nursing operational managers and certain administrative staff such as admissions clerks, an infection control coordinator, and an occupational health coordinator. These posts were filled during the remainder of the operational phase to give a total of 673 personnel.

### 2.4. Qualitative and Quantitative Data Were Derived from Four Sources of Existing Records, Which Were Reviewed to Answer the Research Question

A.A voluntary and anonymous exit survey was administered to all staff on leaving the facility as it was decommissioned, including quantitative and qualitative responses. The staff exit survey was developed by the researchers and made available online and all staff members were encouraged to complete it (Appendix A).B.Electronic records were used for care coordination and transfers between the acute hospitals and the HoH to describe the overall progression of the wave.C.Minutes of all management meetings held during the operational phase of the HoH, including a close-out meeting held at the end of the 10-week period during which managers and clinicians from the acute hospitals gave feedback to the HoH operational management team.D.The COVID-19 infection rates from occupational health records of all staff.

In addition, as the fifth source of data, key informant interviews were held with purposively selected individuals including HoH team leaders, the HoH management team, managers of other health facilities in the Cape Town metropole, as well as head office managers who planned the HoH, using a semi-structured interview guide developed by the research team (Appendix B). Team leaders and managers were specifically selected because of their overview of operations, and the depth of their unique perspectives would not have been gathered by the other sources of data.

In terms of data collection, the records were readily available to the research team. For the key informant interviews, a maximum of 13 potential participants were purposively selected and invited to be interviewed by virtue of their involvement in the HoH, in which all 13 participated. These interviews were held by telephone, digital communication (such as Zoom [16]) and in-person, and were semi-structured in nature. Recordings were professionally transcribed.

Concerning data analysis, quantitative data from the records were entered into Microsoft Excel Spreadsheets [17] and simple descriptive statistics were used to analyze the data and to generate tables and figures. Responses to open-ended questions in the exit survey as well as the minutes of management meetings and key informant interviews were analyzed by thematic analysis using the framework method [18] to produce major and minor themes concerning the research question. The five sets of data including quantitative and qualitative elements were considered together to build a comprehensive picture of each of the emergent themes, as outlined in the results.

In terms of reflexivity, the research team consisted of members of the HoH operational management team (SR, TA, RO, TR, KvP), a HoH clinician (NH) and a research officer employed at the UCT Faculty of Health Sciences deanery (MN). SR, TR and KvP are academic family medicine specialists employed by the University of Cape Town, RO is a family medicine specialist, TA is a physical rehabilitation therapist and health service manager, and NH and MN are both early career medical doctors.

### 2.5. Ethical Considerations

Exit surveys were anonymous at the level of data entry. Electronic data from the referral records, staff leave records and management meeting minutes were rendered anonymous during the process of data collation and analysis. Confidentiality of participants in interviews was maintained by excluding names and identifying details from the transcripts and substituting participant codes, and informed consent was obtained from each one. The protocol was approved by the Human Research Ethics Committee at the University of Cape Town (reference 503/2020), and permission to conduct the study was obtained from the Western Cape Department of Health through the National Health Research Database (reference WC_202010_037).

## 3. Results

The staff exit survey was completed by 190 out of 558 staff members giving a response rate of around 30%. There was a good response rate from all professional staff except for the nursing staff (see Table 1). The survey was completed during the last week before the hospital was closed, by which stage many nurses had been relocated to other work, which may account for the low response rate from the nursing staff. Around two-thirds (31 out of 51) of all medical officers completed the survey. All 11 medical specialists (emergency, internal medicine and family medicine) completed the survey. The length of experience among the respondents ranged from less than 1 year to over 15 years. A quarter of the respondents were not from the Western Cape. The results, however, reflect that there were no major differences in responses given irrespective of place of origin.

In addition to the qualitative data from the exit survey, 13 key informants were interviewed. Table 2 shows the demographic information of these informants.

The findings and main themes of the mixed methods’ sources were organized according to the key aspects of the logic model, as illustrated in Figure 1, consisting of inputs, processes and outcomes. The diagram illustrates the main themes and sub-themes and their inter-relationships as they emerged from the data. Data from the different sources have been combined to present as complete a picture as possible within the framework of Figure 1. The blue box contains the overall aim of the Hospital of Hope as articulated by the management team.

### 3.1. Inputs: Design and Set-Up

It was clear from the interviews with managers that a significant level of leadership was demonstrated in the early phases of the pandemic in March 2020, when high-stakes decisions had to be made quickly, and significant resources were committed in an extremely uncertain context. Putting together a dependable management team in a hurry was initially daunting for the senior manager involved, but he relied on previous experience:

“To be honest the first impression I had was this is not a do-able task and when you are faced with an undoable task you turn to people that have done undo-able tasks for you previously.”(Senior manager)

Pre-existing relationships of trust between managers played a major part in this process, as one put it:

“I must say clearly that the reason we could do that was because we trusted that the system would have our back if something went wrong.”(Senior manager)

The responses of managers and senior clinicians were encouragingly positive, and this spirit of wanting to contribute despite the fear and the risks involved was echoed by the staff who were recruited.

“I think when senior people put up their hands it kind of makes it easier to attract a team around them.”(Manager)

“We were all facing new thing nobody have experience in COVID, from professional nurse to nursing assistant, we were all the same. What a wonderful moment”(Senior nurse)

### 3.2. Process (Implementation)

#### 3.2.1. Person-Centeredness

Many staff members were surprised by the experience of holistic patient care using the family medicine approach, and realized its value in retrospect.

“Having a largely ‘emergency medicine’ background, one often neglects holistic care and focus on a problem-orientated approach to clinical medicine. Working at the CTICC has been an interesting experience in this regard as on more than one occasion, we have seen significant focus on non-clinical components to health care which has invariably and often somewhat surprisingly led to some incredible recoveries.”(Senior clinician)

The medical management of the facility largely by family physicians, however, proved to be justified in retrospect, as a person-centered approach to both staff and patients seemed to make a substantial difference to their respective experiences (see *Outcomes* below).

“I didn’t expect such a primary care family medicine leading orientation, but I think it is working incredibly well, there is no criticism at all but the way the care was managed and the family medicine approach I think is very different to the internal medicine approach.”(Senior clinician)

Staff developed closer relationships with patients than they had expected, due to the nature of the care required, particularly in palliative care.

Thus, video calls with family members at home who were not allowed to visit the patients were considered as equally important as their medical care.

#### 3.2.2. Unity of Purpose

At the onset of the pandemic when HoH was still being commissioned, there was a great deal of fear: fear of the unknown; “fear of missing out” on being on the frontline, combating COVID-19 and preventing illness and death; and fear of contracting and succumbing to COVID-19 itself. However, stronger than that fear was the unity of purpose—a purpose to “contribute to humanity and the country as a whole” and “do the right thing”; a purpose to answer a call during a time of great need; a purpose to learn and grow both personally and professionally; and a purpose to be involved in something new, exciting and possibly “an opportunity of a lifetime”.

This unity of purpose inspired compassion, extreme diligence and hard work, with great personal sacrifice. A culture of comradery and a “can-do” attitude was born even among the staff who were unfamiliar with each other and in some cases had no previous experience.

“When it comes to saving lives and helping people… people will go that extra mile regardless of who they are or where they come from…”(Senior clinician)

“People from all the different angles… [those with] clinical experience, [those at a] managerial level, people that have got only administrative backgrounds, people with certain expertise… just going beyond what fits in their original scope…”(Manager)

There was an overwhelming sense of “togetherness”. It seemed that “at the end of the day everybody was just trying to do their best and everybody was passionate about what they were doing and trying to get the right thing done for their patients.” (Senior clinician)

“Leaving home coming to work made me feel like a hero knowing I’m coming to assist in this difficult time that the country is facing. Helping a patient whenever I was doing my rounds in the wards came easy to me it really woke up the spirit of Ubuntu.”(Clinician)

#### 3.2.3. Teamwork and Communication

The team dynamics were informed by a culture supportive of teamwork, as well as the necessary resources to encourage this style of organizational culture. In terms of leadership, the respondents noted the approachability and transparency of the management teams: “The doors were always open to speak about how we felt at any given time” (Senior clinician). There was also a level of honesty and humility displayed by managers. Managers were “honest about the fact they [they] did not have all the answers” (Manager), but there was a call to collaborate and work together. In turn, this display of honesty and humility may have “put people at ease and they were more willing to come up with innovative ideas and [to] contribute” (Manager). While no explanations could be offered, a sense of trust was quickly established among the different managers despite many not having worked together previously, and the high-stress levels associated with the pandemic and setting up a facility from scratch in such a short space of time. This trust appeared to have increased the level of collaboration and led to lower levels of disappointment and higher levels of forgiveness when expectations were not met, or mistakes were made.

As far as the clinical staff were concerned, a flattening of the hierarchy and the provision of equal opportunities to participate and contribute within teams led to team members feeling valued, respected and validated as team members: “there was true interdisciplinary teamwork” (Senior clinician). This was echoed by a survey respondent: “The environment was very conducive to learning, not hierarchical, completely patient centred and everyone was friendly.”

In the survey, many qualitative responses emphasized the collaborative nature of inter-disciplinary teamwork and connection, and the term “family” was used often. “I didn’t expect such a family experience, all disciplines came together to learn from one another. Mostly the hierarchy fell away.” Other respondents noted equally positive responses, including teamwork, approachable consultants, the positive attitude of all staff, interdisciplinary cohesion, and high emotional support for staff (e.g., wellness area, debriefing). One young pharmacist described experiencing more integration in the team and clinical context, “As a newly registered pharmacist, […] the Hospital of Hope allowed me to experience an interdisciplinary team like never before. Being able to work directly with patients gave me a new sense of passion for my profession by allowing me to see the direct result of my influence.”

One of the key informants noted, that “there were staff who left—they were unhappy, they felt like they were working too hard and that the place was not running well.” (Senior clinician). The survey respondents also described some challenges relating to teamwork and communication. Some of these challenges may relate to the perceived fluid state of organization, especially during the early operational period, with calls, for example, for “earlier designation of team leaders”; “better organization of processes on the ward such as for diabetics, dispensing of meds by nurse”; “better role clarification”; and “more support on the floor with operational issues.”

#### 3.2.4. Staff Safety and Support

A remarkably low level of staff infections with COVID-19 was found. As shown in Table 3, 2.4% of staff tested positive for COVID-19 over a two-month period, compared to an infection rate of 8.1% amongst staff over the same period at a nearby hospital of equivalent size.

Despite initial concerns over personal protective equipment (PPE) and isolated incidents during which PPE was unavailable, overall, staff reported feeling safe. Several reasons were offered by key informants. First, knowing that all patients were COVID-19-positive acted as a constant reminder of the risks of transmission, and maintained a heightened sense of awareness and caution. Second, the high ceilings of the Convention Centre contributed to good ventilation. Except for isolated cases, there was generally adequate PPE; minutes of management meetings record that its proper usage was monitored by dedicated personnel. Thirdly, posters positioned at key points within the hospital served as a constant reminder to use PPE correctly and appropriately, and appropriate occupational health and safety training was performed as well as systems and protocols implemented through daily team huddles, and staff were encouraged to eat well and consume vitamin supplements to boost their immune systems. Stationery and consumables could not be moved from one ward or section of the hospital to another, and mobile phones and devices had to be kept and used within clear plastic bags to prevent fomite transmission. Following the first incident during which there was a shortage of scrubs, management meeting minutes showed that various contingency measures were implemented, including encouraging staff to carry two sets of clothing and the use of jumpsuits.

The survey results reflect uniformity in perceptions of a high level of support provided at HoH amongst all cadres of staff, as reflected in Figure 2. More than 80% of staff felt safe at work, and the majority felt they could access assistance with emotional support when they needed it. Over 80% felt appreciated and valued. The majority also responded that they were able to access assistance with solving problems when the need arose. Only the responses about the debriefing sessions were found to vary amongst the staff, with pharmacists finding the debriefing sessions to be less helpful than the other cadres of staff. The debriefing sessions were also found to be more helpful amongst the more experienced staff (>11 years of work) than the less experienced staff (<1 year).

In qualitative feedback, respondents reflected that morale was high, despite the experience of establishing HoH as a “time of very hard work… trying to put something in place in such a short period, something that didn’t exist and then that was then established in such a short time” (Manager). Additionally, staff absenteeism was at an absolute minimum with “very few staff staying away from work unless they were really, really sick…” (Manager). Complaints about staff, although present, were also scant. Moreover, managers reported that teams did not “require micro-management… [One could] just stand back and let them do the work” (Manager).

It also appeared that staff felt supported, not just physically and with their work but emotionally too. Working at the HoH was “quite an emotional rollercoaster” with high levels of stress for some, but overall, it appeared that staff felt “protected” and supported, and that this may have further contributed to “just giving [their] all”. It was reported that key members of management were often seen on the floor supporting the staff and expressing concern over their needs and emotions, with “check-ins” occurring daily in most cases. One manager remarked that “because [he was] showing [his] face down there, that also gave them the assurance that [the staff] are not left alone that [he was] there with them supporting them as well” (Manager).

Staff support initiatives seemed to have also contributed to the feeling of being supported and to the high levels of morale. These included the various staff wellness initiatives, such as having a senior person on the clinical management team dedicated to staff wellness who was available to staff to discuss any issues they may have had and who co-ordinated the various staff wellness initiatives, such as debriefing sessions; community volunteers distributing gift packs to staff; the “Handwashing” song called “Hands of Hope” (see https://youtu.be/7d_nQkNdM1Q, accessed on 23 March 2023); a staff “Wellness hub”; and a staff appreciation ceremony. Other wellness initiatives included singing and recording the “Jerusalema” dance challenge as a team (see https://www.facebook.com/watch/?v=228267391738155, accessed on 23 March 2023), a common sight during the pandemic at many South African hospitals; and “It really made the place feel like a family!” (Clinician)

In the survey, staff members felt more protected by the PPE than they expected, as evidenced by one staff member saying that: “I was expecting to see some of us contracting the virus, but the government saw to it that we are protected.” Staff members reported that their fears and expectations of danger related to working with COVID-19 patients changed during their experience at HoH. One staff member commented that “At first before coming to work at CTICC I was scared to come because I thought I’m standing a chance of getting infected by the COVID-19 but as time goes on, I felt safe and free at work, and I just became very grateful to be working at CTICC.”

Some survey respondents offered critique: “There was no support at all we were ignored by management because we’re working nights shift, they don’t care about night staff”; “There could’ve been more support on the floor with operational issues. Like where to find stock, office allocations and such.” Negative responses such as these were remarkably few in number.

#### 3.2.5. Responsiveness and Flexibility

Despite the many challenges faced, including the rapid evolution and unfolding of events, a lack of clear reporting lines, roles and responsibilities, and people with differing levels of experience, a novel degree of flexibility and adaptability was displayed in the way Hospital of Hope was able to respond and adapt to the needs of the broader health platform during the first wave of the pandemic. Not only was the Hospital of Hope able to “relieve pressure from the system… and support the system” (Manager), but there was also a consensus by other institutions that its opening and role in the system was “like the cavalry coming over the hill to save them”. (Manager). A senior clinician at the close-out meeting noted that “ [it was the] responsiveness to the need that stood out for me.”

It is also possible that the flexibility and responsiveness were due to the rapid pace of events. “Everything just happened so fast… there was almost no time to think about it too much. I think we were all on this rollercoaster ride.” (Manager). One respondent remarked that “We could have really played hardball and just said listen here, we are not up for this and these are the boundaries and we are reinforcing these boundaries” (Manager). However, a very different approach had been adopted, which was echoed during the closeout meeting: “Responsiveness… There was some criteria for admission… [It was] very quickly established that the criteria was not going to meet the needs [of the rest of the platform]… [There was a] rapid adjustment based on feedback [to meet the needs].”

### 3.3. Outcomes: Achieving the Overall HoH Aim

The key themes related to the implementation and factors that contributed to person-centeredness were described by one key informant as “hitting the target bull’s eye”, a summary that served to organize the data inductively. At its core is the person-centeredness approach, which infused the daily activities linked to patient care and staff wellness.

“So reflecting on that, I think we really hit the target bullseye when we were able to deliver on those patient experiences and giving a good patient experience and I think some of the best comments that came through is where people started talking about hope and just their interaction with the staff and being able to be accommodated in that space and where it was really in the peak of the pandemic.”(Manager)

A number of factors contributed to this sense of achievement, including high job satisfaction, professional and personal development, and discovering a deeper sense of meaning in the work. Finally, a significant outcome was the learning arising from this experience that can be taken into routine health services once the pandemic is over.

#### 3.3.1. High Job Satisfaction

Staff were surprised by the positive nature of their experience, how much they enjoyed working there and how sad they were when it was over. There was an expectation that it would be busier and more stressful than it turned out to be. Staff developed closer relationships with patients and their families than they had expected, and many responses emphasized the nature of interdisciplinary teamwork and connection.

“Despite all the stresses and challenges relating to the COVID response… During this COVID time, I’ve been more creative and productive than I was before then... there are aspects in my job that I… forgot about that I’m rediscovering and enjoying a lot,”(Manager)

Staff also felt more protected by the PPE than they expected they would be.

“I was expecting to see some of us contracting the virus, but the government saw to it that we are protected.”(Nurse)

“I thought I would be affected too but because of the proper PPE I have no fear and I thank you for that,”(Clinician)

The Dedication video includes various staff members reflecting on their experiences (see https://youtu.be/bQCOVNdoYjM, accessed on 23 March 2023) and illustrates this theme in more detail.

#### 3.3.2. Professional Development

Generally, staff felt that the HOH experience would assist their future careers and that their competence improved, as shown by the exit survey results reflected in Figure 3.

An average of over 90% of staff felt more able to do their work after working at HoH, and an overwhelming majority found working at HoH to be beneficial to their future careers. The areas where most learning was found to be achieved were, in descending order, daily work (84%), discussions with peers (64%), in-service training (47%) and then meetings or huddles (46%).

When asked to explain briefly to what extent this experience would assist their future careers, they spoke about leadership skills, self-confidence, experience in teamwork, the value of the multi-disciplinary team, better communication skills and networking, as well as clinical competence.

#### 3.3.3. Personal Development

Most staff enjoyed their work at HoH: on average, across categories of workers, 92% of the survey respondents responded positively to this question (see Figure 4). More than 90% felt they grew personally through their experience at HoH, and less than 40% found going to work stressful. Over 90% found working at HoH meaningful and worthwhile, and 90% of staff reported that their teams worked well together.

In the qualitative responses, some expressed their personal growth as follows:

“I’ve learned a lot, I know things I’ve never known before, met the most wonderful people and I’ve never known what I want for myself but now I know what I want.”

“It was so much more than what I was expecting. In learning about the pandemic and growing as a person, finding my strengths.”

One manager commented on how “working on the frontline with the constant reminder of COVID […] contributed to the development of gratitude and investment in personal relationships with loved ones.” Others expressed how they learned to be more compassionate, and more self-reflective, and how they learned to become better listeners and more effective team players, especially “when there is such a big hype and when things are moving so rapidly”, (Manager).

#### 3.3.4. Meaning

Reflecting on the staff experience, a senior manager highlighted the outcome of hope, which formed an explicit part of the aim as well as the name of the HOH.

“There was a sense of hopelessness descending on the system and when the CTICC started accepting patients that sense of hopelessness or that dark cloud was certainly lifted”.

The expressed intention of the HOH appears to have been vindicated in retrospect, as expressed by staff, the patients and their relatives:

“They experienced it as a time of hope and not a time of despair and I think that was remarkable. Looking back and knowing that we could have contributed in that way to someone’s experience in that time of fear and anxiety and uncertainty, I think was remarkable and that is something personally that one can really reflect on.”(Senior manager)

#### 3.3.5. Implications for the Routine Delivery of Healthcare

Despite its short lifespan, HoH offered numerous learning opportunities and “aha moments” for individuals and the system going forward.

Considering the challenges of the first wave of COVID-19, one respondent provided a novel view of approaching challenges and making mistakes, remarking that,

“Whatever we were doing, we were going to make mistakes… I was going to make mistakes and it was okay, but if I see it as a learning curve it’s going to take away that negative thinking of making a mistake and being hard on myself, but rather trying to see it as it is a learning curve: I must learn from it and then move forward.”(Manager)

The HoH demonstrated the importance of moving towards an integrated, collaborative approach in which institutions adapt and respond to the needs of each other within the system, away from the norm in which each healthcare facility functions in isolation largely oblivious to the needs of the other facilities with the system. To use the words of one manager, “We always say that the Department of Health is not an organization… It is multiple organizations within a massive system and just being able to almost be a bit of a bridge or a glue between some of those various organizations showed me so much that I wouldn’t have learnt in my entire career… [This experience provided] a deeper understanding of how everything is knit together.” Further, another respondent emphasized that the provision of healthcare “only works as a system when it’s a collaborative effort like this was and when it’s system-wide where the needs are met across the entire platform.” (Senior clinician).

During the close-out management meeting, the way the daily huddles were conducted at the HoH was valued. It was cited as a crucial source of information, an effective tool to “identify where the problems are and where the pressures are within the platform” and a means by which that information can be communicated promptly and widely. It was hoped that this can be taken forward into routine services as well.

In discussing the way forward and ideas for the routine health systems in the future, one manager expressed it most clearly as follows:

“… if we can focus that attention just as we did with this pandemic on something like preventative medicine and health promotion, if some of that effort and focus goes into something like this, then I think we will really see an overhaul of our health system. So, the first thing is to be able to clearly identify what are really the health demands and then secondly, being able to respond to that focusing resources, focusing attention, putting people together to really make a difference in those spaces.”

The same manager expressed the need to prioritize clearly in routine service delivery:

“So, to almost get rid of all the clutter. I think the clutter is taking up so much time and really distracting us from what is important, and we have really seen now that what is important is where the health system and the communities meet and what is important in the community in the health sector or from a health perspective must also be what is important for us as a health department and we got this right during this pandemic.”(Manager)

## 4. Discussion

The results reveal surprisingly positive outcomes from the perspective of the staff involved in the field hospital in the first wave, in contrast to the reports from overseas [19,20] as well as in South Africa [21], which described stressed and overworked health personnel struggling to keep up with the overwhelming number of patients. Recent research supports these anecdotal reports of increased incidence of burn-out amongst healthcare workers experienced during the COVID-19 pandemic [22,23,24,25,26].

As reported above, one key informant described this positive situation as “hitting the target bull’s eye” (Figure 5), a summary statement around which we organized the major themes arising from the data in the analysis. In an industrial type of setting such as the Convention Centre with systems established to cope with large numbers of patients, the tendency towards depersonalization was countered by the deliberate focus on person-centeredness, as promoted by the principles of family medicine [27,28] and depicted as the center of the target.

The unity of purpose, frequent communication and ongoing support, together with a culture of learning, is depicted as the next layer from the center. This in turn contributed to better teamwork and the ability to adapt as the pandemic wave changed, shown in the third concentric circle. All of these factors were supported by the necessary resources being mobilized and made available within short timeframes, seen in the outer circle.

The CTICC HOH was conceived, planned and implemented in an extraordinarily short space of time of 12 weeks before the first patients arrived on the 8 June 2020. In contrast to the precedents of the Nightingale Hospitals in the United Kingdom [14] and other temporary facilities established in other countries [4,5,6,7,8,9,10], the crucial strategic decisions were made surprisingly well in retrospect, considering the degree of uncertainty in modeling at the beginning of the first wave. The managers of the Western Cape Provincial Health department demonstrated extraordinary foresight and leadership in mobilizing the necessary resources and making the right calls on major decisions, such as oxygen supply and the number of beds, and a stepwise approach proved wise. However, the results of the study cannot be attributed to resources alone.

As our key informants clarified, the deliberate design of the hospital infrastructure to prioritize infection control by controlling staff and patient flow played an important role in reducing cross-infections and creating efficiencies, in addition to the high roof and air-conditioning of the conference halls, which aided ventilation. The design of the facility appropriately involved infectious disease and emergency medicine consultants with field hospital experience, who worked at great speed to create systems and flows from scratch. When the first patient arrived, however, the medical management was deliberately handed over to family physicians, as appropriate to the intermediate level of care (c.f. Results 2.a. Person-centeredness). This ensured that a person-centered approach pervaded both staff relations and patient care, appropriate to the intermediate level of care, in contrast to the acute level of care where an emergency approach was more appropriate. In addition, the responsiveness of the CTICC HoH to the actual needs of the acute care facilities in the Cape Town metropole, through daily communication across the platform, contributed considerably to the gratitude expressed by the managers, who described the CTICC HoH opening as “just in time” to relieve the load [28].

The central focus on relationships extended to staff as well as patients. The motivation of most staff members who were willing to put themselves and their families at risk, in order to play a part in making a difference to the pandemic, carried significant momentum throughout the period of operation of the HOH, and brought staff together around the shared purpose. This enabled an unusual degree of teamwork and cooperation, as attested to by the pharmacists and physiotherapists, for example. Wong et al. (2020) described the pandemic as a “double-edged sword”, as the camaraderie of close-knit groups of doctors and nurses working together under extreme scenarios can bring out the best in each other, which aligns with the results of this study [29].

Several initiatives to mitigate the effect of the pandemic on staff have been described in the literature, including an emphasis on self-care and connecting with others, as well as systems for support [29] A framework proposed for creating wellness during the pandemic includes the creation of a culture of wellness, increased promotion and access to mental health services, and the use of data and research to enhance support [30].

The study was limited by the response rate to the survey, particularly of nurses, who were under-represented because many were transferred to other hospitals in the last weeks of the HoH. Nurses’ views may have differed from those of other staff categories, but the relatively few survey responses aligned with the qualitative data. The small number of key informant interviews with managers only, by contrast, was by intention. The anonymity of these managers was difficult to guarantee, but as public servants, they were in fact not concerned about being quoted. There were surprisingly few negative comments and little negative feedback in the survey, which may indicate a positive response bias, as those who were unhappy may have left the hospital or have been less likely to complete the survey if they stayed. The exit questionnaire was not piloted before implementation. However, this was mitigated by the multiple sources and types of information collected, including qualitative and quantitative data from documents and interviews.

### Recommendations

The COVID-19 pandemic has shown the world that fundamental changes in health systems cannot wait. COVID-19 revealed structural weaknesses in health systems worldwide, and negatively impacted individuals, societies, and economies. Countries need to take transformative action to build stronger, more resilient health systems that can better prevent, prepare for, and respond to future shocks, while maintaining essential health services [31,32].

Our study findings complement the solutions cited in the literature on how to prevent burnout and improve the mental health of healthcare workers during pandemics, as well as routine healthcare. Traditionally, there has been a large focus on individual strategies to reduce burnout and promote resilience; however, increasing data suggest that such strategies have little to no effect, especially when significant issues are present. A combined approach of giving individuals tools to help them cope, while also providing a system that supports and nurtures, has been recommended in the literature [29].

Based on our findings, we suggest the following recommendations for the future delivery of healthcare:Even outside of pandemic situations, health institutions should move away from the siloed approach of operating independently to a modus operandi of “supporting the greater platform” and adapting rapidly to the changing needs of the healthcare system as a whole;Clinical managers should be allowed a greater level of freedom, flexibility and adaptability in how they run their institutions to alleviate the slower decision-making process and bureaucracy present in the current hierarchical structure of government health departments;The fundamental motivations of applicants during recruitment processes should be probed to ascertain their level of commitment;As echoed by Dutch [33] and Australian [34] studies, healthcare workers should be better supported emotionally, mentally and physically through formal, targeted and dedicated employee wellness and support programs that require mandatory participation on the part of all healthcare workers;A working environment should be created that supports camaraderie among healthcare workers, as well as relationships between healthcare workers and patients;A person-centered approach to healthcare should be adopted throughout the health system, from primary healthcare centers and local clinics up to tertiary-level institutions.

## 5. Conclusions

The staff experience of the first wave at the CTICC HoH was unique and unprecedented, and it produced noteworthy lessons for the design and management of routine health services outside of a disaster situation. The staff recruited to the HoH largely volunteered out of a sense of wanting to contribute positively to mitigating the effects of the pandemic, despite the known risks, which has direct implications for recruitment processes in routine services. Cultivating a unity of purpose amongst the staff is the primary task of every leader in optimizing the impact of an organization. That this unity of purpose was focused on person-centeredness was a deliberate design decision for the intermediate level of care, by appointing family physicians as clinical team leaders. Similarly, the low staff infection rate demonstrated what can be achieved through intelligent design and careful implementation to assure staff of their safety and support. The adaptability and responsiveness of the facility and its staff were largely a product of the unprecedented nature of the pandemic, but such approaches could benefit routine health services enormously, as individual hospitals and health facilities realize their place in a system is “more than the sum of its parts”.

## Figures and Tables

**Figure 1 healthcare-11-00981-f001:**
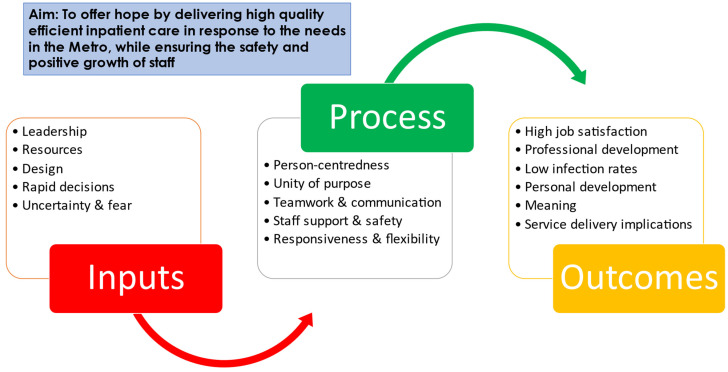
Overview of themes.

**Figure 2 healthcare-11-00981-f002:**
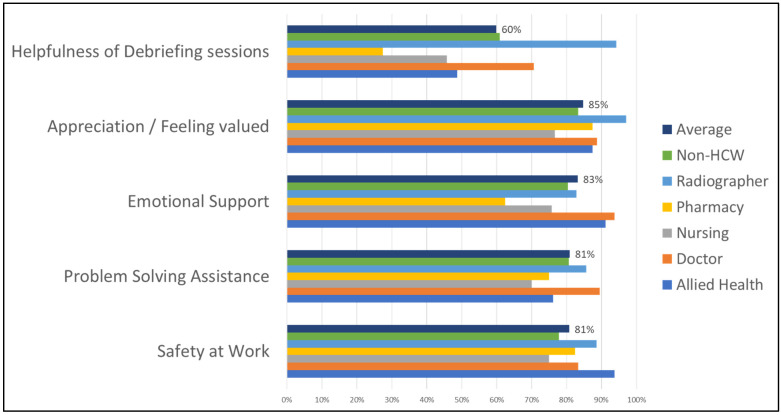
Level of support score per staff category.

**Figure 3 healthcare-11-00981-f003:**
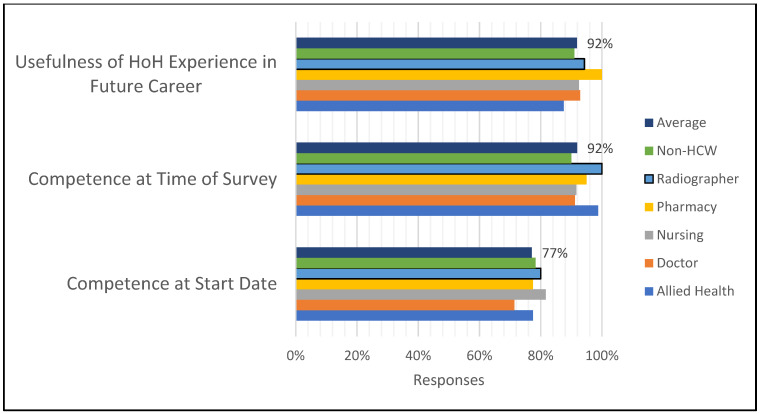
Reported professional development of staff.

**Figure 4 healthcare-11-00981-f004:**
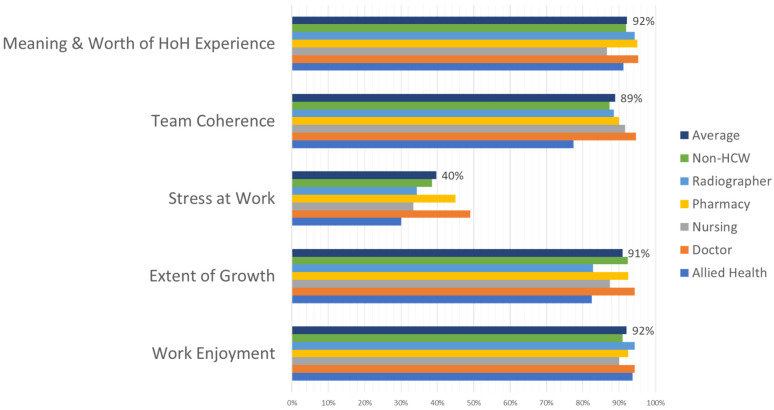
Personal growth score per staff category.

**Figure 5 healthcare-11-00981-f005:**
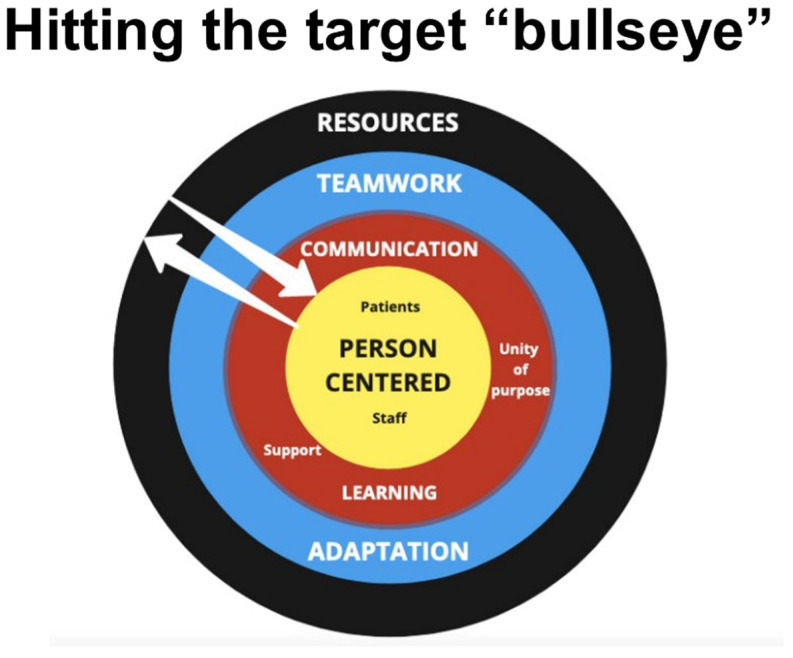
A diagrammatic representation of the main themes.

**Table 1 healthcare-11-00981-t001:** Survey responses by staffing categories.

	Responses	Total Staff	Percentage
Medical Officers	31	51	61
Specialists	11	11	100
Physiotherapists	10	11	91
Social Workers	3	5	60
Dieticians	3	5	60
Radiographers	7	7	100
Pharmacy Staff	8	18	44
Nursing Staff	24	355	7
Non-healthcare workers *	93	95	98
**Total**	**190**	**558**	**34**

* Non-healthcare workers include managers, administrative and other staff.

**Table 2 healthcare-11-00981-t002:** Details of key informants used for qualitative data.

Category	Variable	Value
Gender	Male:Female ratio	7:6
HoH staffing category	Clinical operational manager (nursing, allied health)	5
	Facility management (operational phase)	3
	Facility management (commissioning phase)	3
	External Metro Health Services management	1
Professional background	Administrative/management	2
	Allied health	1
	Medical doctor/specialist	4
	Nursing	5
	Pharmacy	1

**Table 3 healthcare-11-00981-t003:** Staff infections with COVID-19.

Variable	CTICC HoH	Comparable Hospital (Source: Western Cape Department of Health)
Total number of staff on 31 July 2020	673	795
Total staff COVID-19 positive between 8 June and 14 August 2020	16	64
Health professionals who tested COVID-19 positive	16	41
Non-professional staff who tested COVID-19 positive	0	23
**Total staff positivity rate**	**2.4%**	**8.1%**

## Data Availability

The datasets used and/or analyzed during the current study are available from the corresponding author upon reasonable request.

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
