# Peer review of "Tackling the First COVID-19 Wave at the Cape Town Hospital of Hope: Why Was It Such a Positive Experience for Staff?"

_healthcare, 2023, doi:10.3390/healthcare11070981_

Round 1

Reviewer 1 Report (Previous Reviewer 1)

The study reports interesting data on the impact of working in a COVID-19 field hospital on health care workers. Findings from the study will be useful in contributing to the discussion around the implications of working in frontline services during the pandemic for the healthcare workforce. There are a few suggestions to improve the manuscript.

Introduction:

-There are some references missing from the introduction (e.g., after the quotation in the last sentence of paragraph 1 of the introduction, or in paragraph 3 of the introduction following the sentence that ends "...the World Health Organisation (WHO) identified health care workers (HCWs) as being at particular risk of developing burnout and fatigue because of working with COVID-19 patients". Referencing throughout the introduction and discussion should be checked carefully.

Ethical considerations

- The study has received ethical approval. It would be interesting to know if there were any specific ethical issues associated with maintaining anonymity given the relatively small sample size for the interviews (n=13), particularly in light of the sampling frame. The challenges around this as well as the implications for study findings (particularly given the study team comprised senior colleagues of interviewees) should be discussed in the limitations section.

Results

- Formatting: The way quotations are formatted and presented is inconsistent throughout the results section. I would recommend using the first approach (i.e., quotes presented as a new paragraph indented from main text, with participant identifier included if available) for readability. 

- As this is a mixed methods study the integration of data/findings from each phase is important. It is not always clear how the different data points feed into the analytic framework - it appears as if not all data sources are relevant to all of the themes and maybe this needs to be acknowledged in either the methods or results section for clarity. In particular it is not clear how findings from the documentary analysis have been used. 

- The use of a conceptual map is helpful (figure 1) but the bullet points in this figure do not always map onto the subheadings in text e.g., in "3. Outcomes: Achieving the overall HoH aim" (note there is a numbering error in the manuscript), there is a point "e. Implications for the routine delivery of healthcare" that is not contained in the map. 

Also, the reporting in subsection "1. Inputs: Design and set-up, is very brief compared to the other subsections and formatting is different compared to other subsections which also include subheadings so it is not immediately clear if all bullet points from the map have been addressed. 

- In the staff and safety section, paragraph two, it is unclear which data is being referred to in this summary. If the interviews or free text responses of the survey could an illustrative quote be added? If from the documentary analysis could this be made clear?

Discussion:

- The diagrammatic representation of 'hitting the target bull's eye' (figure 5) is introduced in the discussion but it is not really discussed in the results or methods despite apparently framing the themes arising from the data. Greater clarity is required about how and when this framework was used and where it comes from (i.e., it is not clear if this arises from the data or if it is based on prior theory, and how this was used in analysis of data - did it inform the Framework Method adopted?). An explanation of how this relates to the conceptual map presented earlier is also required. Presented as it is the relevance of this diagram is unclear and it creates confusion in the discussion. It may be that this needs to be explained and presented earlier in the manuscript. 

- Paragraphs 4 and 5 contains a number of statements that either need to be referenced, or if they come from the data (the documentary analysis or meeting minutes perhaps?) need to be clearly reported in the results section. 

Author Response

Reviewer 2 Report (New Reviewer)

Dear authors,

The paper discusses a very interesting and specific topic of emergencies in a serious pandemic. Yet, this issue now is not so relevant, because the critical phase, was two years ago. After that time this investigation and report would be more profitable. Minor issues- English language and style are fine and minor spell check is required. The discussion and all the paper were developed in a way too extensive. It should be changed to a smaller version, more interesting to the readers.

Author Response

Manuscript submitted to MDPI Healthcare: ID healthcare-2208720

Title: Tackling the first COVID-19 wave at the Cape Town Hospital of Hope: Why was it such a positive experience for staff?

Tabulated responses to Round 2 reviewer’s comments:

REVIEWER 2

The paper discusses a very interesting and specific topic of emergencies in a serious pandemic. Yet, this issue now is not so relevant, because the critical phase, was two years ago. After that time this investigation and report would be more profitable. Minor issues- English language and style are fine and minor spell check is required.

Respectfully, we disagree - we explain the relevance to current practice which is not restricted to a pandemic situation. In fact this was the whole motivation for writing the article in the first place.

The discussion and all the paper were developed in a way too extensive. It should be changed to a smaller version, more interesting to the readers.

A previous reviewer in round 1 requested that we extend the discussion, which was felt to be too concise. No change has been made.

This manuscript is a resubmission of an earlier submission. The following is a list of the peer review reports and author responses from that submission.

Round 1

Reviewer 1 Report

Background

This study adds to a growing body of literature reporting on the impact of working during COVID-19 on the healthcare workforce. More could be done to contextualise the study within the International literature (both related specifically to COVID and related to workforce development more generally). Limited references to the wider literature are currently provided in the background section. Further literature on the impact of working during the pandemic on frontline staff, comparative literature from other pandemics, and literature looking at International trends in recruitment and retention, staff satisfaction and burnout would be useful.

Methods

I would recommend further information in the methods section.

(1) Firstly, there seem to be 5 sets of data used for the study however it is not clear how all of these have been used to answer the research question.

(2) It would be useful to have a brief summary of the data collected in the exit survey (I understand that there is an appendix 1 included but I could not access this and in any case a brief summary in text would be useful).

(3) It is not clear how the Vula Mobile and electronic records have been used.

(4) Presumably the key informant interviews should be listed as data set E? Again I could not access the Appendix 2 for this but a brief summary of the content of the topic guide would be useful.

(5) Also, it is reported that the key informant interviews were purposively sampled. However, only managers and senior staff appear to have been sampled here. Is this intentional or was recruitment of other staff not possible? This should be highlighted as a limitation. 

(6) More information is required on the quantitative data analysis. I understand that Excel was used for descriptive purposes but the results refer to "significant differences". Further information on statistical tests used is needed here. 

(7) More information is required on data synthesis. Although not explicitly stated this appears to be a mixed methods study (mixed methods sources are referred to in the results section) but important detail is missing on *how* and *when* synthesis occurred. 

(8) Ethical considerations are listed however informed consent is not mentioned. Further information is required. Also, anonymisation of data is referred to. Whilst this might be possible for survey data I would have thought it unlikely that meeting minutes could be anonymised due to the nature of most qualitative data sets. Suggest using the term pseudonymised or deidentified for clarity. 

Results

The results section required more detail:

(1) The data on response rate is useful but only partially presented. What is a "good" response rate in this context and what was the response rate for nursing staff? When you say 100% of consultants/two thirds of medical doctors completed the survey what number does this refer to and what proportion of the overall respondents does this represent?

(2) I would recommend summarising the survey data in a table. In particular, a more detailed presentation of participant characteristics (including percentages and n, and any measures of central tendency for continuous data). Were any additional demographic data collected?

(3) Paragraph 1 of the results section refers to Cuban doctors – does this need to be explained?

(4) In the last sentence of paragraph 1 of the results section “no significant differences” with regard to responses and place of origin are referred to. How was this tested? More data needs to be presented.

(5) The characteristics of interview participants have not been described. A summary table of demographic data relating to this sample should be presented.

(6) Little information describing the other datasets has been provided.

(7) The use of a figure to present the key themes is very useful. Some more description in text of this logic model would be useful. How does the aim listed in the blue box related to the themes outlined?  

(8) It would be useful if you could provide more specific identifiers for your illustrative quotes so that we can see that they come from a range of participants i.e., other than just job role. Also, the quotes from survey respondents have no identifiers. It is it possible to provide some based on the demographic data provided in the surveys?

(9) On page 5 you refer to CTICC1 and CTICC2 – how are these related to CTICC/CTICC HoH – to make this more understandable to an international audience you need to explain the local context.

(10) Your illustrative quotes contain a lot of specific detail i.e., they have not been deidentified in places – from an ethical perspective does this maintain the confidentiality of your participants given the small sample size?

(11) Some of your findings do not seem to be supported by the data as it is presented. For example, on page 8 you discuss staff perception of safety yet provide no data to support some of the claims made. Also, you refer to Figure 3 – could values be provided for all staff categories? Also, you refer to variation by levels of experience but no data is provided?

(12) The presentation of illustrative quotes is inconsistent and often very dense. Recommend choosing fewer quotes overall and presenting them in a consistent way.

Discussion and conclusion

As with your background your discussion could bring in relevant International literature to support. Your findings appear to echo some of the positive outcomes reported elsewhere from studies looking at the experiences healthcare staff working through covid, so it would be good to place them in this context. However, taking into account some of the limitations of this study I think it is crucial to discuss the impact on the conclusions that can be drawn from your results. Your discussion is also very brief, particularly in comparison to your results section which is very descriptive. Recommend structuring your discussion so that it is in line with the themes identified in your findings.

In the limitations section you highlight the sampling (and the under-representation of nurses in particular) as a potential issue. I would recommend further discussion of this point and how it may impact your findings and conclusions. Also, you make reference to the healthcare workers who left the HoH – could this explain the more positive experiences in your data/the absence of more critical perspectives? Similarly, many of your illustrative quotes in the discussion appear to be from senior managers/managers/senior staff – was this group and their experiences potentially overrepresented in your sample?

You mention some implications for practice and policy in your conclusions but recommend making this more explicit and as above supporting with literature.

Reviewer 2 Report

The introduction needs to be expanded and connected to the scientific literature to a much larger degree. This version of the manuscript reads more like a compilation of empirical data than a scientific article. The introduction does not inform the reader why this study is important in terms of a knowledge gap, nor how it relates to previous research. There is no connection to theory.

More details regarding how data were analyzed are needed.

The various types of empirical data are interesting, but could be condensed and analyzed in a more focused manner. At this point, the results section constitutes 2/3 (!) of the entire manuscript.

To make the manuscript more balanced, the discussion should be expanded and the results should be analyzed more in depth in relation to previous research to a larger degree than now.
